# VideoGuide: Improving Video Diffusion Models without Training Through a Teacher's Guide

| Base | Ours | Base | Ours |

"A drone view of celebration with Christmas tree and fireworks"     "A boat sailing in the middle of the ocean"

| Base | Ours | Base | Ours |

"Slow motion footage of a racing car"     "A dog drinking water"

Figure 1: VideoGuide is a novel framework for improving temporal consistency while preserving imaging quality, enabling high-quality video generation for diverse text prompts. By applying VideoGuide to underperforming base models, we can significantly improve temporal consistency with no additional training or fine-tuning. *Best viewed with Acrobat Reader. Click each image to play the video clip.*

## Abstract

Text-to-image (T2I) diffusion models have revolutionized visual content creation, but extending these capabilities to text-to-video (T2V) generation remains a challenge, particularly in preserving temporal consistency. Existing methods that aim to improve consistency often cause trade-offs such as reduced imaging quality and impractical computational time. To address these issues we introduce VideoGuide, a novel framework that enhances the temporal consistency of pretrained T2V models without the need for additional training or fine-tuning. Instead, VideoGuide leverages *any* pretrained video diffusion model (VDM) or itself as a guide during the early stages of inference, improving temporal quality by interpolating the guiding model's denoised samples into the sampling model's denoising process. The proposed method brings about significant improvement in temporal consistency and image fidelity, providing a cost-effective and practical solution that synergizes the strengths of various video diffusion models. Furthermore, we demonstrate prior distillation, revealing that base models can achieve enhanced text coherence by utilizing the superior data prior of the guiding model through the proposed method. Project Page: https://videoguide2025.github.io/

# 1 INTRODUCTION

Text-to-image (T2I) diffusion models have greatly changed the way how visual content is created and distributed, enabling users to effortlessly generate creative images from detailed text descriptions. Now the AI community is looking deeper into the potential of T2I diffusion models, exploring their application to the higher dimensional field of video generation. Text-to-video (T2V) diffusion models aim to extend the capabilities of their image-based counterparts by generating coherent video sequences from text descriptions, handling both spatial and temporal dimensions simultaneously. However T2V diffusion models still show sub-standard performance regarding temporal consistency, and can lead to the generation of degraded samples. Poor temporal consistency is also the main challenge for a variety of tasks, such as creation of personalized T2V models. Hence, recent work (Wu et al., 2023; Qiu et al., 2023; Ge et al., 2024) aims to enhance various aspects of temporal quality, but suffers from problems such as degraded quality, slow inference, etc. In this work, we attend to the clear absence of a reliable method for refining the temporal quality of pretrained text-to-video (T2V) generation models, and propose a novel framework for improved generation that does not require any training or fine-tuning.

Specifically, we introduce VideoGuide, a general framework that uses any pretrained *video* diffusion model as a *guide* during early steps of reverse diffusion sampling. Choice of the pretrained VDM is flexible: it is either identical to the VDM used for inference, or it is freely selected from all existing VDMs. In any case, the VDM that acts as the guide provides a consistent video trajectory by proceeding in its own denoising for a small number of steps. The guiding model's denoised sample is then integrated into the original denoising process to guide the sample towards a direction with better temporal quality. Through interpolation, the sampling VDM is able to follow the temporal consistency of the guiding VDM to produce samples of enhanced quality. Such interpolation only needs to be involved in the first few steps of inference, but is strong enough to guide the entire denoising process towards more desirable results. Remarkably, interpolating information of the guiding model's denoised sample has the effects of providing the base model a better noise prior, even guiding the model to create samples that were previously unreachable. VideoGuide is a versatile framework in that any pretrained video diffusion model can be used for distillation in a plug-and-play fashion. By incorporating a superior VDM as a video guide, our framework can be used to boost underperforming VDMs into state-of-the-art quality. This is particularly useful when the relatively underperforming VDM possesses unique traits unavailable for the superior VDM.

In particular, we show two representative cases of how VideoGuide can be applied, with Animate-Diff (Guo et al., 2024) and LaVie (Wang et al., 2023). In AnimateDiff, a motion module is trained that can be interleaved into any pretrained T2I model. The scheme works for any personalized image diffusion model and grants easy application of controllable and extensible modules (Zhang et al.; Guo et al., 2023), but not without consequences. Specifically, fixing the T2I weights limits interaction between the temporal module and generated spatial features, hence harming temporal consistency. Applying VideoGuide with an open-source state-of-the-art model without personalization capability (Chen et al., 2024) as the guiding model, we can greatly enhance the temporal quality of AnimateDiff. This allows us to combine the best of both worlds: personalization and controllability is provided by the base model, while temporal consistency is refined by the guiding model. Likewise, LaVie is a multifaceted T2V model that offers various functions including interpolation and super-resolution in a cascaded generation framework, but shows substandard temporal consistency. Using VideoGuide, we can upgrade its temporal consistency with an external model while maintaining its multiple functions.

The synergistic effects that our framework can bring are not limited to these two cases but are, in fact, boundless. As powerful video diffusion models emerge, existing models will not become obsolete but actually improve through the guidance our method provides. Moreover, as VideoGuide can be applied solely during inference time, these benefits can be enjoyed with no cost at all. Our contributions can be summarized as follows:

1. We propose VideoGuide, a novel framework for enhancing temporal consistency and motion smoothness while maintaining the imaging quality of the original VDM.

2. We show how *any* existing VDM can be incorporated into our framework, enabling boosted performance of inadequate models along with newfound synergistic effects among models.

3. We provide evidence of prior distillation, in which the informative prior of guidance models can be utilized to create samples of improved text coherency.

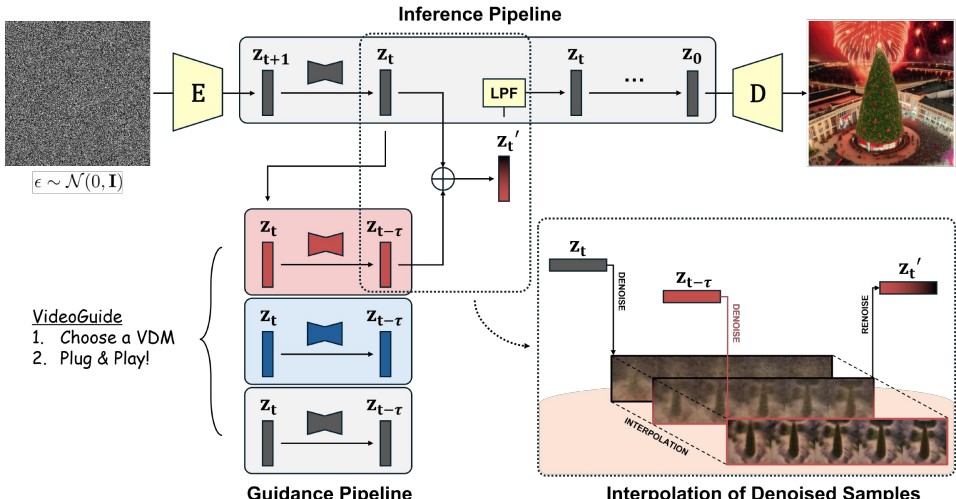

Figure 2: **Overall Pipeline.** VideoGuide is a framework for enhancing temporal quality without additional training, leveraging the capabilities of any pretrained VDM. Throughout the denoising process of the sampling VDM, the guiding VDM receives an intermediate latent $z_t$ and provides a temporally consistent sample $z_{t-\tau}$ by proceeding in its own denoising for a small number of steps $\tau$. The sample $z_{t-\tau}$ is then denoised and interpolated with the denoised $z_t$ to produce a fused latent $z'_t$. Such interpolation only needs to take part in the first few steps of inference, and effectively guides samples towards a direction of improved temporal consistency. To further ensure model flexibility in refining high-frequency areas for better image quality, the latent $z'_t$ is passed through a Low-Pass Filter (LPF). Overall, VideoGuide is a straightforward addition to the original pipeline, yet it is powerful enough to significantly enhance temporal consistency without compromising imaging quality or motion smoothness.

## 2   RELATED WORKS

**The Diffusion Model.** Diffusion probabilistic models (Ho et al., 2020) have achieved great success as generative models. To address the significant computational cost that arises from operating in pixel space, Latent Diffusion Models (LDMs) (Rombach et al., 2021) learn the diffusion process in latent space. LDMs utilize an encoder-decoder framework where the encoder $\mathcal{E}$ and the decoder $\mathcal{D}$ are trained together to reconstruct the input data. This training aims to satisfy the relation $\boldsymbol{x} = \mathcal{D}(\boldsymbol{z}_0) = \mathcal{D}(\mathcal{E}(\boldsymbol{x}))$, where $\boldsymbol{z}_0$ is the latent representation of the corresponding clean pixel image $\boldsymbol{x}$. Thus the forward diffusion process in latent space is defined as follows:

$$\boldsymbol{z}_t = \sqrt{\bar{\alpha}_t}\boldsymbol{z}_0 + \sqrt{1 - \bar{\alpha}_t}\boldsymbol{\epsilon}, \tag{1}$$

where $\bar{\alpha}_t$ is a pre-determined noise scheduling coefficient, and $\boldsymbol{\epsilon} \sim \mathcal{N}(0, \mathbf{I})$ represents Gaussian noise sampled from a standard normal distribution. The reverse diffusion process is directed by a score-based neural network, denoted as the diffusion model $\boldsymbol{\epsilon}_\theta$, which is trained using the denoising score matching framework (Ho et al., 2020; Song et al., 2021b). The training objective for this model is formulated as follows:

$$\min_\theta \mathbb{E}_{t, \boldsymbol{\epsilon} \sim \mathcal{N}(0, \mathbf{I})} ||\boldsymbol{\epsilon} - \boldsymbol{\epsilon}_\theta(\boldsymbol{z}_t, t)||_2^2. \tag{2}$$

Following the formulation of DDIM (Song et al., 2021a), the reverse deterministic sampling from the posterior distribution $p(\boldsymbol{z}_{t-1}|\boldsymbol{z}_t, \boldsymbol{z}_0)$ is given by:

$$\boldsymbol{z}_{t-1} = \sqrt{\bar{\alpha}_{t-1}}\boldsymbol{z}_{0|t} + \sqrt{1 - \bar{\alpha}_{t-1}}\boldsymbol{\epsilon}_\theta(\boldsymbol{z}_t, t) \tag{3}$$

$$\boldsymbol{z}_{0|t} = \frac{\boldsymbol{z}_t - \sqrt{1 - \bar{\alpha}_t}\boldsymbol{\epsilon}_\theta(\boldsymbol{z}_t, t)}{\sqrt{\bar{\alpha}_t}} \tag{4}$$

where the denoised sample at timestep $t$, denoted as $\boldsymbol{z}_{0|t}$, can be obtained using Tweedie's formula.

**Classifier Free Guidance (CFG).** In conditional diffusion models, classifier free guidance (Ho & Salimans, 2021) enhances quality of generated samples by increasing the conditional likelihood through a weighted adjustment of the conditional distribution. Mathematically this is expressed as:

$$\hat{\epsilon}_\theta(z_t, t) = \epsilon_\theta(z_t, t, \phi) + w[\epsilon_\theta(z_t, t, c) - \epsilon_\theta(z_t, t, \phi)] \tag{5}$$

where $c$ and $\phi$ refer to the text condition and null condition, respectively, and $w$ refers to the guidance scale used during reverse sampling. To apply classifier free guidance to Eq. (3) and Eq. (4), we substitute $\epsilon_\theta(z_t, t)$ with $\hat{\epsilon}_\theta(z_t, t)$ in both. Recent work (Chung et al., 2024) points out that using a high guidance scale $w$ (*e.g.*, around 7.5) often results in issues such as abrupt changes and color saturation in the denoised sample $z_{0|t}$ during the early timesteps of reverse sampling. To address these issues, CFG++ (Chung et al., 2024) introduces interpolation between the conditional estimate $\epsilon_\theta(z_t, t, c)$ and the unconditional estimate $\epsilon_\theta(z_t, t, \phi)$ using a lower guidance scale $w \in [0, 1]$. Derived from score distillation sampling (SDS) (Poole et al., 2022), CFG++ replaces the renoising term $\hat{\epsilon}_\theta(z_t, t)$ into $\epsilon_\theta(z_t, t, \phi)$. In this case, Eq. (3) can be modified as below:

$$z_{t-1} = \sqrt{\bar{\alpha}_{t-1}} z_{0|t} + \sqrt{1 - \bar{\alpha}_{t-1}} \epsilon_\theta(z_t, t, \phi) \tag{6}$$

Our proposed interpolation scheme operates on denoised samples for early timesteps, during which maintaining high-quality denoised samples is essential. Thus, we utilize CFG++ throughout the early stages of denoising to achieve smooth interpolation.

**Video Diffusion Model (VDM) & Consistent Video Generation.** The Video Diffusion Model (VDM), originally proposed in Ho et al. (2022), operates the diffusion process in the video domain. Similar to LDMs, many recent VDMs (Xing et al., 2023; Chen et al., 2023; He et al., 2022) are trained in the latent space to reduce computational cost. In Latent VDMs (LVDMs), a temporal layer is incorporated to facilitate frame interaction along the temporal axis during training. By modifying $z_t$ to $z_t^{1:N}$ in Eqs. (1)-(6), the diffusion model can be extended to the video domain. For simplicity, we will use the notation $z_t$ to represent the latent for video generation instead of $z_t^{1:N}$.

One of the main challenges in utilizing diffusion models for video generation lies in maintaining temporal consistency. In the video domain, PYoCo (Ge et al., 2024) introduces a carefully designed progressive video noise prior to better leverage image diffusion models for video generation. However, PYoCo primarily focuses on the noise distribution during the training stage and requires extensive fine-tuning on video datasets. Recent work (Qiu et al., 2023; Jiaxi et al., 2023) also attempts to improve temporal consistency, but focuses more on long video generation and is not applicable to the basic 16 frame scenario. FreeInit (Wu et al., 2023) addresses the issue of video consistency by iterative refinement of the initial noise. This method aims to resolve the training-inference discrepancy in video diffusion models by reinitializing noise with a spatio-temporal filter, ensuring the refined noise better aligns with the training distribution. While this approach enhances frame-to-frame consistency, it has a significant drawback: repeated iteration leads to the loss of fine details and imaging quality degradation. Additionally, the iterative nature of the method induces high computational costs, prolonging the generation process.

In VideoGuide, we aim to enhance video consistency without the aforementioned drawbacks. By integrating a small number of guidance steps into the original reverse sampling process, we are able to avoid image degradation while significantly reducing inference time compared to prior work. Furthermore, our method can incorporate external diffusion models to facilitate more temporally consistent video generation. This makes our approach particularly effective for models that struggle with temporal consistency but demonstrate strong performance in other areas (*e.g.*, customizable T2I-based video models (Guo et al., 2024)).

## 3 VIDEOGUIDE

### 3.1 VIDEO CONSISTENCY ON DIFFUSION TRAJECTORY

The DDIM formulation can be expressed as a proximal optimization problem (Kim et al., 2024):

$$z_{t-1} = \sqrt{\bar{\alpha}_{t-1}} z' + \sqrt{1 - \bar{\alpha}_{t-1}} \hat{\epsilon}_\theta(z_t, t) \quad \text{where} \quad z' = \arg\min_z ||z - z_{0|t}||_2^2 \tag{7}$$

We extend this approach to the video domain by introducing a novel regularization term specially crafted for enhancing temporal consistency.

Specifically, for a given video $x_r^{1:N}$, suppose that a temporally consistent latent of $z_r = \mathcal{E}(x_r^{1:N})$ exists. Then, it would be desirable to set the optimization problem as follows:

$$\min_{z} ||z - z_{0|t}||_2^2 + \lambda_{reg} R(z) \quad \text{where} \quad R(z) := ||z - z_r||_2^2 \tag{8}$$

Unfortunately, it is infeasible to provide $z_r$ as the purpose of the VDM is to generate new *unseen* samples. Thus, we propose to use $z_{0|t-\tau}$ as a proxy of $z_r$ where $\tau$ is a sufficient number of timesteps. This is because $z_{0|t-\tau}$ is usually a cleaner and temporally more consistent sample than $z_{0|t}$, so we want to utilize this property. Under this assumption, the highly complex problem of generating temporally consistent video samples is reduced to solving the simple optimization problem below:

$$z_{t-1} = \sqrt{\bar{\alpha}_{t-1}}z' + \sqrt{1 - \bar{\alpha}_{t-1}}\hat{\epsilon}_\theta(z_t, t)$$
$$\text{where} \quad z' = \min_{z} ||z - z_{0|t}||_2^2 + \lambda_{reg}||z - z_{0|t-\tau}||_2^2 \tag{9}$$

which is equivalent to

$$z_{t-1} = \sqrt{\bar{\alpha}_{t-1}}\left(\beta z_{0|t} + (1-\beta)z_{0|t-\tau}\right) + \sqrt{1 - \bar{\alpha}_{t-1}}\hat{\epsilon}_\theta(z_t, t), \quad \beta = \frac{1}{1 + \lambda_{reg}} \tag{10}$$

Accordingly, it suffices to use the interpolation of $z_{0|t}$ and $z_{0|t-\tau}$ as an estimate of the temporally consistent form of $z_{0|t}$. To further ensure model flexibility to refine high-frequency areas for better image quality, we employ a low-pass filter inspired by previous work (Wu et al., 2023). Specifically, using a low-pass filter and high-pass filter of cut-off frequency $\gamma$, denoted $LPF_\gamma$ and $LPF_\gamma$ respectively, we define the following update:

$$z_{t-1} = LPF_\gamma(z_{t-1}) + HPF_{1-\gamma}(\epsilon) \quad \text{where} \quad \epsilon \sim N(0, \mathbf{I}) \tag{11}$$

Replacement of high-frequency regions with random Gaussian noise enhances model capacity to infer corresponding high-frequency components, leading to denoised results of higher quality.

## 3.2 GUIDANCE WITH EXTERNAL VIDEO DIFFUSION MODELS

The assumption $z_r \approx z_{0|t-\tau}$ in Sec. 3.1 holds for any sample $z_{0|t-\tau}$ with temporal consistency comparable to a real-world sample. This brings us to realize that the sample $z_{0|t-\tau}$ does not necessarily have to originate from the same base model. It is possible to *plug in* any denoised latent $z_{0|t-\tau}$ from any video diffusion model, and the denoising process would be guided to follow the temporal consistency of the supplemented latent. Here, we demonstrate the steps required for utilizing denoised samples $z_{0|t-\tau}^{(G)}$ of an external guidance model $G$ to enhance the performance of the base sampling model $S$.

**Renoising into the Guidance Domain.** Different video diffusion models are trained on different noise schedules and distributions, and matching such discrepancies is a mandatory process. When utilizing a guiding model with conflicting factors, the intermediate latent $z_t$ of the sampling model must be transformed to align with the noise schedule and distribution of the guiding model. The transformation process can be defined as follows:

$$z_t^{(G)} = \sqrt{\bar{\alpha}_t^{(G)}}z_{0|t}^{(S)} + \sqrt{1 - \bar{\alpha}_t^{(G)}}\epsilon, \quad \text{where} \quad \epsilon \sim N(0, \mathbf{I}) \tag{12}$$

where $(S)$ denotes the components related to the base sampling model and $(G)$ denotes the components related to the external guiding model. Specifically, $z_{0|t}^{(S)}$ is the denoised sample from $z_t^{(S)}$ at timestep $t$, and $\bar{\alpha}_t^{(G)}$ is derived from the noise schedule of the guiding diffusion model. The resulting outcome $z_t^{(G)}$ can then be denoised with the guiding model for a sufficient number of timesteps $\tau$ up to $z_{0|t-\tau}^{(G)}$.

**Interpolation of Denoised Samples.** Interpolating the denoised samples of the two models $S$ and $G$ can be expressed as below:

$$z_{t-1}^{(S)} = \sqrt{\bar{\alpha}_{t-1}}(\beta z_{0|t}^{(S)} + (1-\beta)z_{0|t-\tau}^{(G)}) + \sqrt{1 - \bar{\alpha}_{t-1}}\hat{\epsilon}_\theta^{(S)}(z_t, t) \tag{13}$$

Note that the only difference from Eq. (10) is the introduction of the $z_{0|t-\tau}^{(G)}$ term, where originally $z_{0|t-\tau}^{(S)}$ would be used. $LPF_\gamma$ can then be used on $z_{t-1}^{(S)}$ as in Eq. (11) for replacing high-frequency components:

$$z_{t-1}^{(S)} = LPF_\gamma(z_{t-1}^{(S)}) + HPF_{1-\gamma}(\epsilon) \quad \text{where} \quad \epsilon \sim N(0, \mathbf{I}) \tag{14}$$

**Synergistic Effects of External VDM Guidance.** Utilizing a high-performance open-source model (Chen et al., 2024) as the guiding diffusion model in our VideoGuide framework is shown to improve temporal consistency even while achieving faster convergence. Compared to the self-guided case, generating temporally coherent samples from a superior model proves beneficial to the quality of the resulting samples, as illustrated in Sec. 4. Moreover, since interpolation occurs only during the early timesteps, the advantages of the sampling diffusion model—such as the personalized video generation and controllability of AnimateDiff—are fully preserved. Accordingly, VideoGuide is a versatile framework that can combine the best of both worlds: the sampling model and the guiding model. No additional training or fine-tuning is required for seeing such synergistic effects, allowing the user to freely select favored VDMs in a plug-and-play fashion.

### 3.3 VIDEOGUIDE IN PRACTICE

**Early Timestep Interpolation.** In VideoGuide a novel interpolation technique is included in the inference process, and the equations above explain cases at a specific timestep $t$. Theoretically this interpolation could be performed at every denoising timestep, but such iteration would both be computationally expensive and detrimental to the high-frequency components that emerge at later timesteps. Recent work (Wu et al., 2023) shows that providing informative low-frequency components at initialization time is sufficient for enhancing temporal consistency. Likewise, we find that applying our interpolation scheme at early timesteps is adequate for enforcing temporal consistency while allowing high-frequency regions to align more closely to the low-frequency structure. An extensive ablation study regarding the number of interpolation steps is given in Sec. 5.1

**Prior Distillation.** Each video diffusion model spans its own specific data distribution, causing sample generation to be restricted to the data prior the model has been trained on. Thus, if the data prior of a model is substandard, the generation results of the model are also inherently substandard. This is especially noticeable when using personalized text-to-image (T2I) models such as Dreambooth or LoRA in AnimateDiff, in which substandard results that do not align with the given text prompt are frequently observed. Prior work (Ge et al., 2024) elaborates on the importance of data prior for VDMs, but the proposed solution involves extensive fine-tuning, making it impractical for simple use cases. On the other hand, VideoGuide comes as a potential solution in such cases, where the interpolation between two models exhibit a form of prior distillation. Through the guidance of a generalized video diffusion model (*e.g.* Chen et al. (2024)) the base sampling model is able to refer to the denoised sample provided by the guidance model, and steer its sampling process towards a relevant outcome. This allows for the effective generation of diverse objects, even while retaining the style of the original data domain. For the case of AnimateDiff, the approach allows for broader customization without the need for directly training the personalized T2I model on a wider range of data. Extensive analysis concerning this issue is provided in Sec. 5.2.

## 4 EXPERIMENTS

**Experimental Settings.** In our experiments, we leverage multiple open-source Text-to-Video (T2V) diffusion models to explore the combined strengths of each. For the guiding diffusion model, we choose Videocrafter2 (Chen et al., 2024) due to its strong performance in temporal consistency, as measured by the VBench (Huang et al., 2024) benchmark. For sampling, we employ AnimateDiff (Guo et al., 2024) for flexible personalization of video content, and Lavie (Wang et al., 2023) to enhance video quality and increase frame count through super-resolution and interpolation techniques. This integration combines the temporal consistency of the guiding model with the advantages of the sampling model. All experiments were conducted using DDIM with 50 steps for sampling. For our experiments with AnimateDiff, we set $I = 5$, $\beta = 0.5$, and $\tau = 10$, and used the Butterworth filter with a normalized frequency of $0.25$ and a filter order of $n = 4$. Additional experimental details are provided in Appendix A.

| Method | Subject consistency (↑) | Background Consistency (↑) | Imaging Quality (↑) | Motion Smoothness (↑) |
|---|---|---|---|---|
| AnimateDiff (Guo et al., 2024) | 0.9183 | 0.9437 | 0.6647 | 0.9547 |
| AnimateDiff + FreeInit (Wu et al., 2023) | 0.9487 | 0.9604 | 0.6173 | 0.9705 |
| AnimateDiff + Ours (with AnimateDiff) | 0.9596 | 0.9642 | 0.6526 | 0.9760 |
| AnimateDiff + Ours (with VideoCrafter2) | **0.9614** | **0.9664** | **0.6671** | **0.9772** |
| LaVie (Wang et al., 2023) | 0.9534 | 0.9599 | 0.6750 | 0.9658 |
| LaVie + FreeInit (Wu et al., 2023) | 0.9625 | 0.9643 | 0.6533 | **0.9757** |
| LaVie + Ours (with Lavie) | 0.9629 | **0.9652** | 0.6780 | 0.9725 |
| LaVie + Ours (with VideoCrafter2) | **0.9635** | 0.9643 | **0.6796** | 0.9723 |

Table 1: Quantitative comparison of video generation. **Bold**: best, underline: second best.

**Evaluation Metrics.** To validate the improvement in video consistency with our proposed method, we evaluate four key metrics: subject consistency, background consistency, imaging quality, and motion smoothness. For subject consistency evaluation, DINO (Caron et al., 2021) feature similarity between frames is measured to assess consistency of the subject's appearance throughout the video. Background consistency is evaluated using CLIP feature similarity between frames to evaluate overall scene consistency. Imaging quality is also a key metric in that maintaining original image quality is essential for generation and enabling customization. Thus we evaluate image quality using the multi-scale image quality transformer (MUSIQ) (Ke et al., 2021), which measures frame-wise low-level distortion such as noise, blur, and over-exposure. Additionally, to ensure smooth motion, we employ a video interpolation model (Li et al., 2023) to assess consistency of motion across video frames.

## 4.1 COMPARISON RESULTS

Qualitative results for various prompts and base models are shown in Fig. 3. Samples from the base model show impairment in temporal consistency, such as fluctuation in color or abrupt change in subject appearance. FreeInit moderately solves the problem of temporal consistency but at the cost of considerable degradation in imaging quality, such as smoothing out of textural details. In contrast, the proposed VideoGuide significantly enhances temporal consistency without loss of imaging quality or motion smoothness. Furthermore, VideoGuide solves sudden frame shifts frequently observed in LaVie by providing smooth frame transitions, explained in Appendix E. Detailed explanation of base models used and additional qualitative results are included in Appendix A and Appendix E.

In quantitative comparison, our method demonstrates superior performance over the base model, achieving improvements in both subject and background consistency. When using AnimateDiff as the base model, our approach shows best results for all key metrics. There is a notable enhancement in temporal consistency compared to baselines, and such increase is not at the cost of imaging quality or motion smoothness. Our method is shown to actually improve both factors when VideoCrafter2 is used as the guiding model. A small decrease in imaging quality can be observed for the self-guided case, but the difference is minimal compared to the notable decrease in imaging quality for FreeInit. When using LaVie as the base model, our approach still shows a reliable increase in subject and background consistency. Note that increase is relatively smaller due to a higher base consistency. Furthermore, our method successfully maintains imaging quality and improves motion smoothness. Such results conform with our original purpose to create a method for improving temporal consistency while preserving imaging quality and motion smoothness. Additionally we conduct a user study to prove the effectiveness of our approach regarding Text Alignment, Overall Quality, and Smooth And Dynamic Motion, further explained in Appendix C.

Regarding computational efficiency, iterative initial noise refinement in prior work (Wu et al., 2023) requires performing DDIM sampling for multiple iterations, resulting in a high computational cost. In contrast, our method only introduces a small number of additional sampling steps. This difference leads to a significant reduction in inference time, yielding a $\times 1.8 \sim \times 2.5$ improvement in generation speed for AnimateDiff and a $\times 2.1 \sim \times 3.1$ improvement for Lavie as shown in Tab. 2.

| Method | AnimateDiff | LaVie |
|---|---|---|
| FreeInit | 51.88 | 28.18 |
| Ours (self-guided) | **21.02** | **8.99** |
| Ours (VC-guided) | 29.22 | 13.43 |

Table 2: Inference time for video generation(*s*). Both the self-guided case and the VideoCrafter2-guided case show significant decrease in inference time. **Bold**: best, underline: second best.

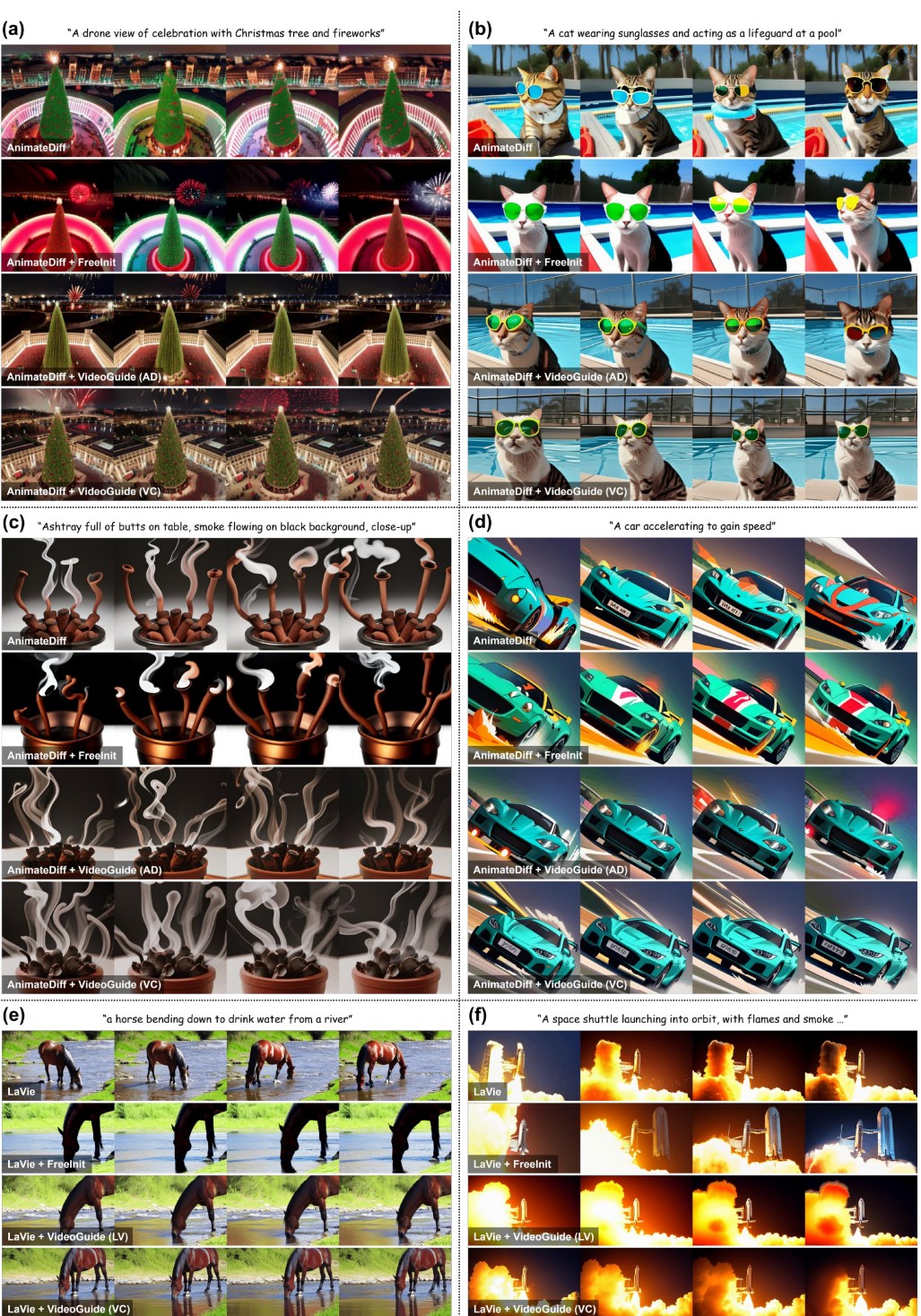

Figure 3: **Qualitative Comparison.** VideoGuide is applied on various base models for different text prompts. For each prompt, frames of generated samples from four different models are displayed: (i) **Base model** (first row); (ii) **Base model with FreeInit** (second row); (iii) **Base model with VideoGuide (self-guided case)** (third row); (iv) **Base model with VideoGuide (external model-guided case)** (fourth row). AD, VC, LV indicate guidance models of AnimateDiff, VideoCrafter-2.0, LaVie, respectively. Samples for the base model show substandard temporal consistency, especially regarding color fluctuation and subject appearance change. Applying FreeInit improves consistency but introduces degradation in imaging quality, such as smoothing out of textural details. In contrast, applying VideoGuide significantly enhances temporal consistency while preserving imaging quality, both for the self-guided case and the external model-guided case.

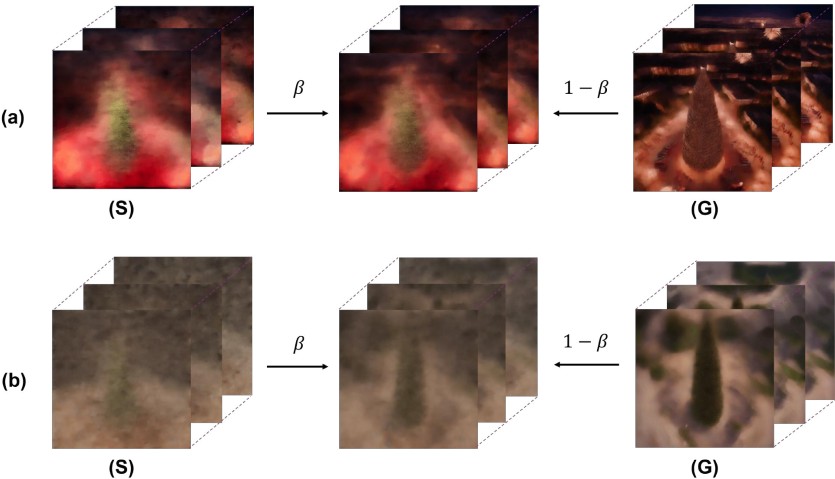

Figure 4: (a) The interpolation process between denoised samples from the sampling model (S) and the guiding model (G) for high guidance scale $w = 7.5$ is shown. (b) The interpolation process for low guidance scale $w = 0.8$ is shown. Both interpolations are performed at $T = 980$ and $\beta = 0.7$. Results indicate that with high guidance scale $w$, influence of the guiding diffusion model is significantly reduced due to color saturation.

| Interpolation Scale $\beta$ | | | Interpolation Step Number $I$ | | | Guidance Step Number $\tau$ | | |
|---|---|---|---|---|---|---|---|---|
| | SC | BC | | SC | BC | | SC | BC |
| $\beta = 0.9$ | 0.9518 | 0.9599 | $I = 1$ | 0.9524 | 0.9618 | $\tau = 1$ | 0.9444 | 0.9558 |
| 0.8 | 0.9546 | 0.9609 | 2 | 0.9489 | 0.9588 | 3 | 0.9531 | 0.9611 |
| 0.7 | 0.9576 | 0.9628 | 3 | 0.9546 | 0.9612 | 5 | 0.9582 | 0.9641 |
| 0.6 | 0.9605 | 0.9649 | 4 | 0.9602 | 0.9645 | 7 | 0.9611 | 0.9658 |
| 0.5 | **0.9614** | **0.9664** | 5 | **0.9614** | **0.9664** | 10 | **0.9614** | **0.9664** |

Table 3: Ablation study regarding interpolation scale $\beta$, number of interpolation steps $I$, and number of guidance sampling steps $\tau$. Subject consistency (SC) and background consistency (BC) is compared for various parameters. **Bold**: best, underline: second best.

## 5 ANALYSIS

### 5.1 ABLATION STUDY

**Importance of Guidance Scale $w$.** Recent study (Chung et al., 2024) demonstrates that employing a high CFG scale ($w > 1.0$) in the early timesteps of diffusion sampling leads to off-manifold behavior. This phenomenon results in denoised samples exhibiting problems such as color saturation and abrupt transitions, which negatively affect the interpolation between samples during these timesteps. We solve this by applying a lower guidance scale $w$ during the early stages of sampling, ensuring smoother interpolation between the denoised samples. As illustrated in Fig. 4 (a), when using a high CFG scale ($w = 7.5$), the influence of the guiding diffusion model becomes minimal due to significant color saturation, making it difficult for the output of the guiding model to be reflected effectively. In contrast, as illustrated in Fig. 4 (b), a lower CFG scale ($w = 0.8$) facilitates smoother interpolation between the sampling diffusion model and the guiding diffusion model. This highlights the importance of clean interpolation in our method, as improper guidance can lead to sub-optimal performance. Further analysis about CFG and CFG++ can be found in Appendix B.

**Parameter Selection.** An analysis is performed to assess how varying parameters of the guiding diffusion model impacts temporal consistency. Specifically, we examine the effects of three factors: interpolation scale $\beta$, number of interpolation steps $I$, and number of guidance sampling steps $\tau$.

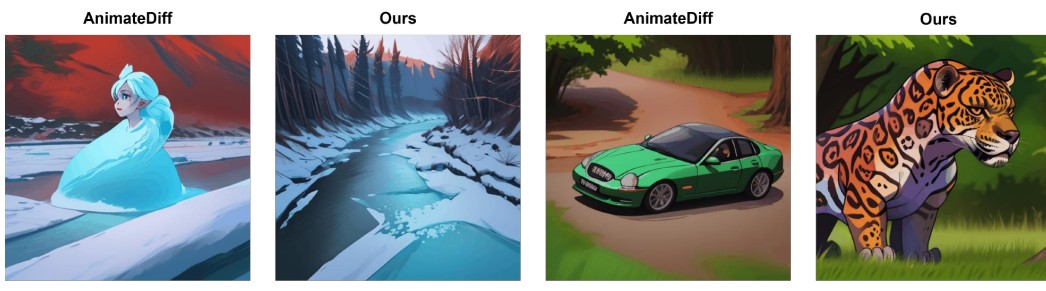

| AnimateDiff | Ours | AnimateDiff | Ours |

"A footage of a frozen river"  "A jaguar is in the park"

Figure 5: **Prior Distillation Results.** VideoGuide solves degraded performance regarding text coherency by enabling the utilization of a superior data prior. Example results for certain ambiguous prompts are displayed. For each prompt, the same random seed is shared for both methods. Animate-Diff directs generation of 'beetle' and 'jaguar' towards car samples due to a substandard data prior. Using VideoGuide, users can distill a superior prior for correct generation.

Temporal consistency is evaluated for both Subject Consistency (SC) and Background Consistency (BC). To secure efficient sampling time, we limit maximum values to $\tau = 10$ and $I = 5$.

Our ablation studies prove that all three parameters are closely related to temporal consistency. Decrease in interpolation scale $\beta$, which is analogous to increase in the influence of the guiding diffusion model, leads to improved subject and background consistency. Note that the minimum value of $\beta$ is constrained to $0.5$ to mitigate the risk of distribution shift. Increasing the number of interpolation steps $I$ also leads to improvement in temporal consistency, which proves that our interpolation scheme is indeed effective. Furthermore, increasing the number of guidance sampling steps $\tau$ enhances consistency, indicating that blending intermediate latents with better-denoised versions enhances overall consistency as expected (*i.e.*, $z_{0|t-\tau} \approx z_r$). Such ablation study highlights the trade-off between consistency improvement and computational efficiency, offering insight into optimal parameter settings for the guiding diffusion model.

## 5.2 PRIOR DISTILLATION

Degraded performance due to a substandard data prior is an issue only solvable through extra training. However VideoGuide provides a workaround to this matter by enabling the utilization of a superior data prior. Fig. 5 demonstrates example cases. For all instances, generated samples are guided towards a result of better text coherence while maintaining the style of the original data domain. Additional examples of prior distillation are provided in Appendix E.

## 6 CONCLUSION

In this work, we introduced VideoGuide, a novel and versatile framework that enhances the temporal quality of pretrained text-to-video (T2V) diffusion models without the need for additional training or fine-tuning. Our approach provides temporally consistent samples to intermediate latents during the early stages of the denoising process, guiding the low frequency components of latents towards a direction of high temporal consistency. The samples provided are not confined to the base model; any superior pretrained VDM can be selected for distillation. By doing so, we empower underperforming models with improved motion smoothness and temporal consistency while maintaining their unique traits and strengths, including personalization and controllability. We demonstrate the effectiveness of VideoGuide on various base models, and prove its ability to enhance temporal consistency without sacrifice of imaging quality or motion smoothness compared to prior methods. The potential of VideoGuide extends far beyond the cases discussed, as VideoGuide ensures that even existing models can remain relevant and competitive by leveraging the strengths of superior models. As video diffusion models continue to evolve, new and emerging VDMs will only enhance the pertinence of VideoGuide over time, broadening the scope of VDMs utilizable as a video guide.

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

Lvmin Zhang, Anyi Rao, and Maneesh Agrawala. Adding conditional control to text-to-image diffusion models.

## A  EXPERIMENTAL DETAILS

### A.1  PROMPT SELECTION

In all experiments, we utilize 800 prompts from various categories in VBench (Huang et al., 2024) to evaluate the model's ability to generate across diverse categories.

### A.2  HYPERPARAMETER SELECTION

We employ a classifier-free guidance (CFG) scale of 7.5 during inference for both base models (AnimateDiff, LaVie) and FreeInit-applied cases. During interpolation of the denoised samples, we apply CFG++ reverse sampling with a guidance scale of $w = 0.8$ in DDIM 50-step sampling. After completing the interpolation step, we revert to CFG reverse sampling with a CFG scale of 7.5. In FreeInit, we use a Butterworth filter with a normalized frequency of 0.25, filter order $n = 4$, and perform 5 iterations, as recommended in prior work. The same filter is applied in our experiments with FreeInit. For AnimateDiff, we configure the guiding model with parameters $I = 5$, $\beta = 0.5$, and $\tau = 10$. In the case of LaVie, we set $I = 3$, $\beta = 0.5$, and $\tau = 10$ to optimize inference speed. Additionally, the $\tau$ intervals are not uniformly spaced as in the standard 50-step DDIM sampling. To better leverage temporally consistent samples, we divide the remaining interval into 25 steps for reverse sampling during guidance steps. Also, we found that applying renoising to guidance sampling is more effective in improving consistency in the case of self-guidance. Therefore, we incorporated renoising during self-guidance in a similar manner as when using a external model for guidance.

### A.3  FIGURE EXPLANATION

Base models used for **Figure 3**:
(a) AnimateDiff with pretrained T2I model RealisticVision.
(b) AnimateDiff with pretrained T2I model RealisticVision.
(c) AnimateDiff with pretrained T2I model ToonYou.
(d) AnimateDiff with pretrained T2I model FilmVelvia.
(e) LaVie.
(f) LaVie.
Base model used for **Figure 5**: AnimateDiff with pretrained T2I model ToonYou.

## B  QUANTITATIVE ANALYSIS OF CFG AND CFG++

There may be concerns that the effectiveness of our method in improving consistency stems from the use of the CFG++ algorithm itself. To address this, we provide results for using CFG and CFG++ across the Base Model, Base Model + FreeInit, and Base Model + VideoGuide. The results demonstrate that CFG++ is particularly effective for interpolation. As shown in Tab. 4, metrics for Base and FreeInit decrease when CFG++ is used, and metrics improve only when CFG++ is applied to our interpolation scheme. This implies the significant positive impact on consistency of CFG++ within the proposed interpolation scheme, especially compared to CFG. Also, this supports the idea, as discussed earlier in Sec. 5.1, that smooth interpolation of denoised samples positively impacts model performance.

| Metrics | Base | | FreeInit | | Ours | |
|---|---|---|---|---|---|---|
| | CFG | CFG++ | CFG | CFG++ | CFG Interp. | CFG++ Interp. |
| Subject Consistency (↑) | 0.9183 | 0.9176 | 0.9487 | 0.9473 | 0.9598 | 0.9614 |
| Background Consistency (↑) | 0.9437 | 0.9435 | 0.9604 | 0.9604 | 0.9635 | 0.9664 |

Table 4: Comparison of consistency metrics between CFG and CFG++ in AnimateDiff. Results indicate that interpolating denoised samples with CFG++ has a larger impact on improving both subject and background consistency.

## C    USER STUDY

We conduct a user study to evaluate generated video samples using three criteria: **Text Alignment**, **Overall Quality**, and **Smooth And Dynamic Motion**, with all metrics scored on a 1 to 5 scale. A total of 30 participants provided ratings for each metric, offering comprehensive feedback on the generated videos.

**Text Alignment**

- Measures how well the video corresponds to the prompt, focusing on semantic coherence.
- Question: Do you think the videos reflect the given text condition well?
  (5: Strongly Agree / 4: Agree / 3: Neutral / 2: Disagree / 1: Strongly Disagree)

**Overall Quality**

- Assesses the video's visual consistency, image degradation, and aesthetic appeal.
- Question: Do you think the video's overall quality is good? (rich detail, unchanging objects)
  (5: Strongly Agree / 4: Agree / 3: Neutral / 2: Disagree / 1: Strongly Disagree)

**Smooth And Dynamic Motion**

- Evaluates the naturalness and fluidity of the motion in the video.
- Question: Do you think the video's overall motion is smooth and dynamic?
  (5: Strongly Agree / 4: Agree / 3: Neutral / 2: Disagree / 1: Strongly Disagree)

| Method | Text Alignment | Overall Quality | Smooth And Dynamic Motion |
|---|---|---|---|
| Base | 3.72 | 2.84 | 2.9 |
| Base + FreeInit | 3.97 | 3.35 | 3.38 |
| Base + VideoGuide (Ours) | **4.36** | **4.37** | **4.36** |

Table 5: User Study. **Bold**: best, underline: second best.

Tab. 5 shows that our method surpasses the baseline and previous work in all evaluated aspects.

## D    PSEUDO CODE

Pseudo codes regarding our algorithm are provided in the following page.

## E    MORE QUALITATIVE EXAMPLES

Additional samples are provided in following pages:

- Supplemental examples of prior distillation.
- Qualitative comparison for various base models.
- Usage of VideoGuide to solve sudden frame shifts in LaVie samples.

---

**Algorithm 1** VideoGuide with Sampling Diffusion Model

---

**Require:** guidance scale $\lambda \in [0, 1]$, guiding steps $I$, interpolation scale $\beta$, extra step $\tau$

1: Initialize $\boldsymbol{z}_T \sim \mathcal{N}(0, \mathbf{I})$
2: **for** $t = T, \ldots, 1$ **do**
3:     $\hat{\boldsymbol{\epsilon}}_\theta(\boldsymbol{z}_t, t) = \boldsymbol{\epsilon}_\theta(\boldsymbol{z}_t, t, \phi) + \lambda[\boldsymbol{\epsilon}_\theta(\boldsymbol{z}_t, t, c) - \boldsymbol{\epsilon}_\theta(\boldsymbol{z}_t, t, \phi)]$
4:     $\boldsymbol{z}_{0|t} = (\boldsymbol{z}_t - \sqrt{1 - \bar{\alpha}_t} \hat{\boldsymbol{\epsilon}}_\theta(\boldsymbol{z}_t, t))/\sqrt{\bar{\alpha}_t}$
5:     $\boldsymbol{z}_t = \sqrt{\bar{\alpha}_t} \boldsymbol{z}_{0|t} + \sqrt{1 - \bar{\alpha}_t} \boldsymbol{\epsilon}, \quad where \quad \boldsymbol{\epsilon} \sim N(0, \mathbf{I})$
6:     **if** $T - t < I$ **then**
7:       **for** $j = 0, \ldots, \tau$ **do**
8:         $\boldsymbol{z}_{t-j-1} = \sqrt{\bar{\alpha}_{t-j-1}} \boldsymbol{z}_{0|t-j} + \sqrt{1 - \bar{\alpha}_{t-j-1}} \boldsymbol{\epsilon}_\theta(\boldsymbol{z}_{t-j}, t - j, \phi)$
9:       **end for**
10:      $\boldsymbol{z}'_{0|t} = \beta \cdot \boldsymbol{z}_{0|t} + (1 - \beta) \cdot \boldsymbol{z}_{0|t-\tau}$
11:      $\boldsymbol{z}_{t-1} = \sqrt{\bar{\alpha}_{t-1}} \boldsymbol{z}'_{0|t} + \sqrt{1 - \bar{\alpha}_{t-1}} \boldsymbol{\epsilon}_\theta(\boldsymbol{z}_t, t, \phi)$
12:      $\boldsymbol{z}_{t-1} = LPF_\gamma(\boldsymbol{z}_{t-1}) + HPF_\gamma(\boldsymbol{\epsilon}), \quad where \quad \boldsymbol{\epsilon} \sim N(0, \mathbf{I})$
13:     **else**
14:      $\boldsymbol{z}_{t-1} = \sqrt{\bar{\alpha}_{t-1}} \boldsymbol{z}_{0|t} + \sqrt{1 - \bar{\alpha}_{t-1}} \boldsymbol{\epsilon}_\theta(\boldsymbol{z}_t, t, \phi)$
15:     **end if**
16: **end for**
17: **Output:** Final video $\boldsymbol{z}_0$

---

**Algorithm 2** VideoGuide with Guiding Diffusion Model

---

**Require:** guidance scale $\lambda \in [0, 1]$, guiding steps $I$, interpolation scale $\beta$, extra step $\tau$, Guiding Model $G$ parameterized by $\psi$, noise schedule $\bar{\alpha}^{(G)}$ of $G$

1: Initialize $\boldsymbol{z}_T \sim \mathcal{N}(0, \mathbf{I})$
2: **for** $t = T, \ldots, 1$ **do**
3:     $\hat{\boldsymbol{\epsilon}}_\theta(\boldsymbol{z}_t, t) = \boldsymbol{\epsilon}_\theta(\boldsymbol{z}_t, t, \phi) + \lambda[\boldsymbol{\epsilon}_\theta(\boldsymbol{z}_t, t, c) - \boldsymbol{\epsilon}_\theta(\boldsymbol{z}_t, t, \phi)]$
4:     $\boldsymbol{z}_{0|t} = (\boldsymbol{z}_t - \sqrt{1 - \bar{\alpha}_t} \hat{\boldsymbol{\epsilon}}_\theta(\boldsymbol{z}_t, t))/\sqrt{\bar{\alpha}_t}$
5:     $\boldsymbol{z}_t^{(G)} = \sqrt{\bar{\alpha}_t^{(G)}} \boldsymbol{z}_{0|t} + \sqrt{1 - \bar{\alpha}_t^{(G)}} \boldsymbol{\epsilon}, \quad where \quad \boldsymbol{\epsilon} \sim N(0, \mathbf{I})$
6:     **if** $T - t < I$ **then**
7:       **for** $j = 0, \ldots, \tau$ **do**
8:         $\boldsymbol{z}_{0|t-j}^{(G)} = (\boldsymbol{z}_{t-j}^{(G)} - \sqrt{1 - \bar{\alpha}_{t-j}^{(G)}} \hat{\boldsymbol{\epsilon}}_\psi(\boldsymbol{z}_{t-j}^{(G)}, t - j))/\sqrt{\bar{\alpha}_{t-j}^{(G)}}$
9:         $\boldsymbol{z}_{t-j-1}^{(G)} = \sqrt{\bar{\alpha}_{t-j-1}^{(G)}} \boldsymbol{z}_{0|t-j}^{(G)} + \sqrt{1 - \bar{\alpha}_{t-j-1}^{(G)}} \boldsymbol{\epsilon}_\psi(\boldsymbol{z}_{t-j}^{(G)}, t - j, \phi)$
10:      **end for**
11:      $\boldsymbol{z}'_{0|t} = \beta \cdot \boldsymbol{z}_{0|t} + (1 - \beta) \cdot \boldsymbol{z}_{0|t-\tau}^{(G)}$
12:      $\boldsymbol{z}_{t-1} = \sqrt{\bar{\alpha}_{t-1}} \boldsymbol{z}'_{0|t} + \sqrt{1 - \bar{\alpha}_{t-1}} \boldsymbol{\epsilon}_\theta(\boldsymbol{z}_t, t, \phi)$
13:      $\boldsymbol{z}_{t-1} = LPF_\gamma(\boldsymbol{z}_{t-1}) + HPF_\gamma(\boldsymbol{\epsilon}), \quad where \quad \boldsymbol{\epsilon} \sim N(0, \mathbf{I})$
14:     **else**
15:      $\boldsymbol{z}_{t-1} = \sqrt{\bar{\alpha}_{t-1}} \boldsymbol{z}_{0|t} + \sqrt{1 - \bar{\alpha}_{t-1}} \boldsymbol{\epsilon}_\theta(\boldsymbol{z}_t, t, \phi)$
16:     **end if**
17: **end for**
18: **Output:** Final video $\boldsymbol{z}_0$

---

**AnimateDiff**                                              **Ours**

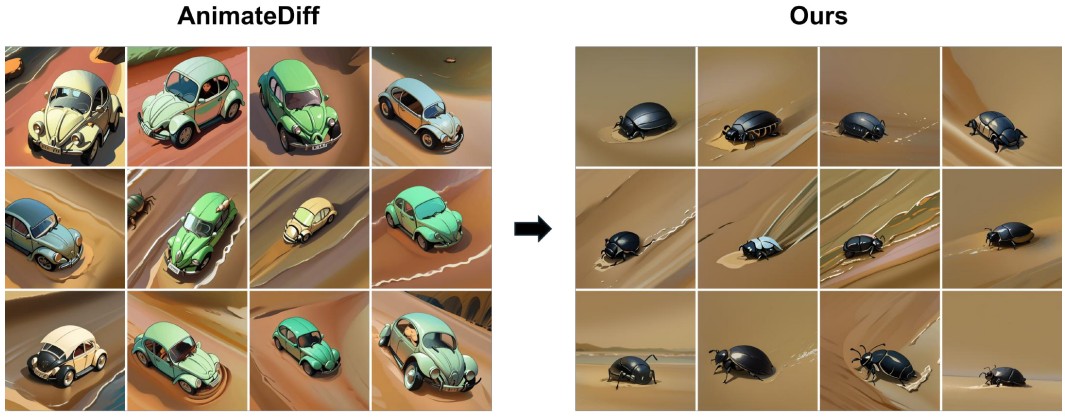

"A beetle is on the sand"

**AnimateDiff**                                              **Ours**

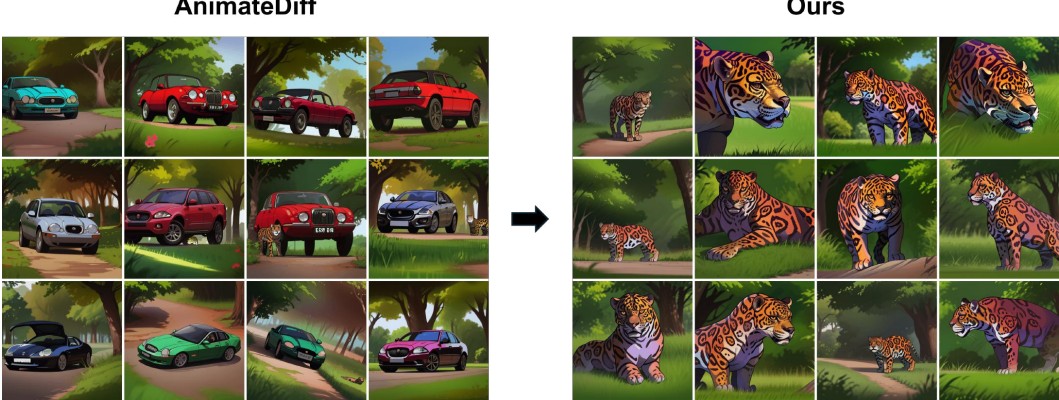

"A jaguar is in the park"

**AnimateDiff**                                              **Ours**

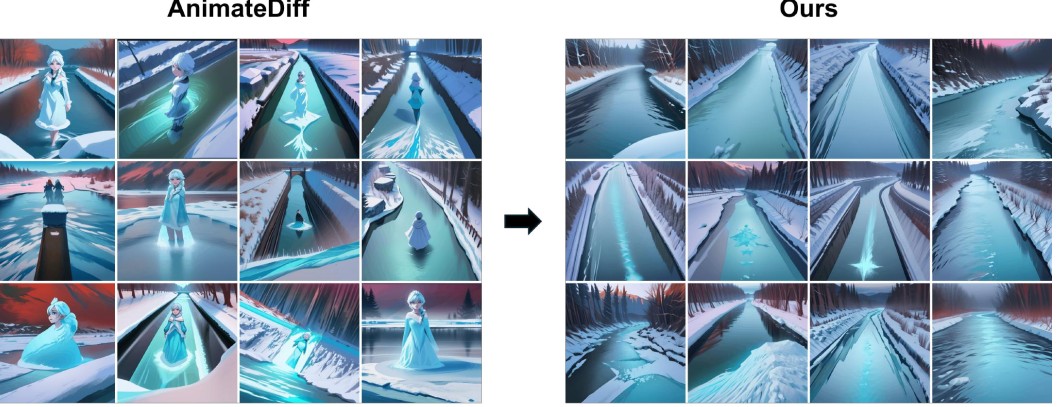

"A footage of a frozen river"

Figure 6: **Prior Distillation**. For each prompt, we share the same random seed for both methods.

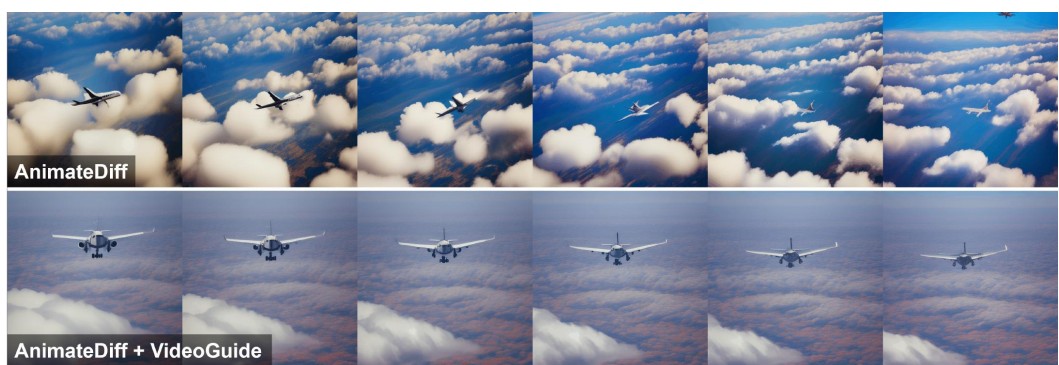

"An airplane flying above the sea of clouds"

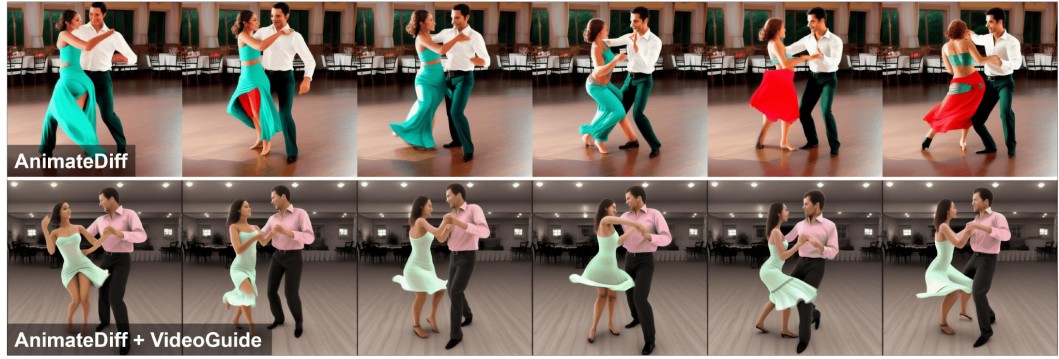

"Couple salsa dancing"

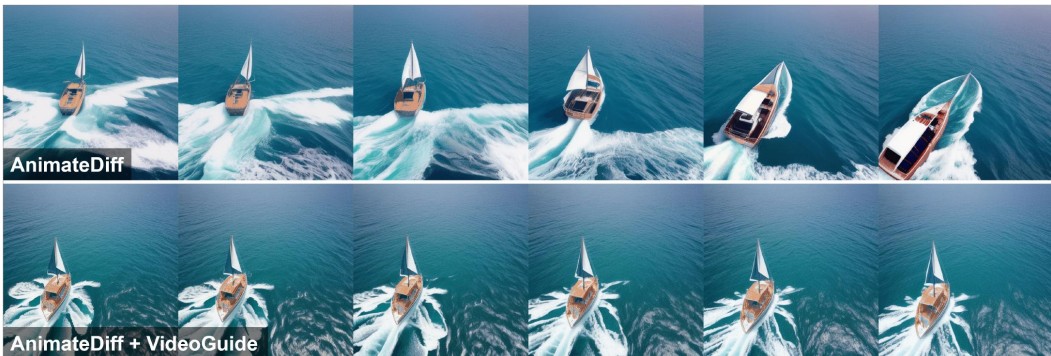

"Boat sailing in the middle of ocean"

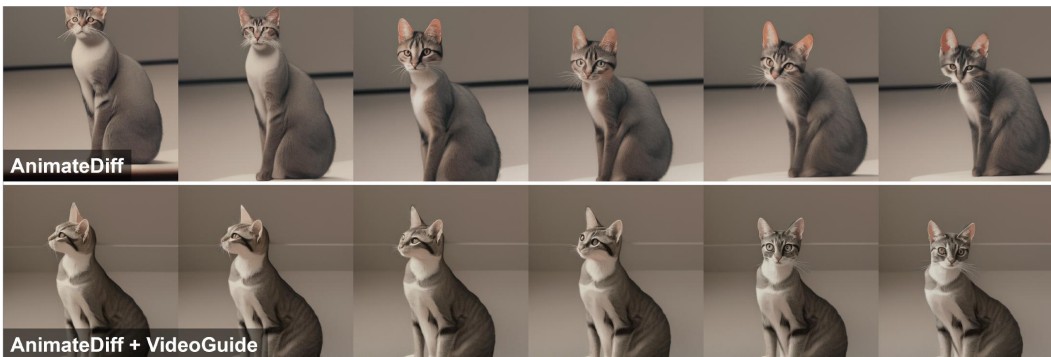

"Curious cat sitting and looking around"

Figure 7: More Qualitative Results of VideoGuide on AnimateDiff (with RealisticVision).

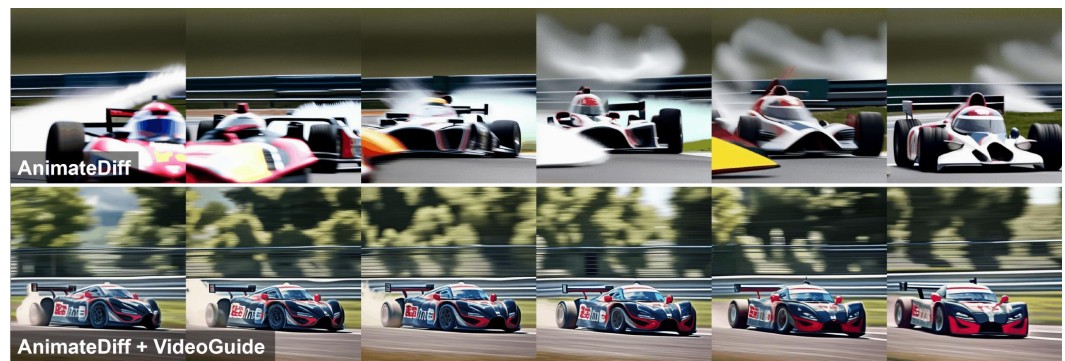

"Slow motion footage of a racing car"

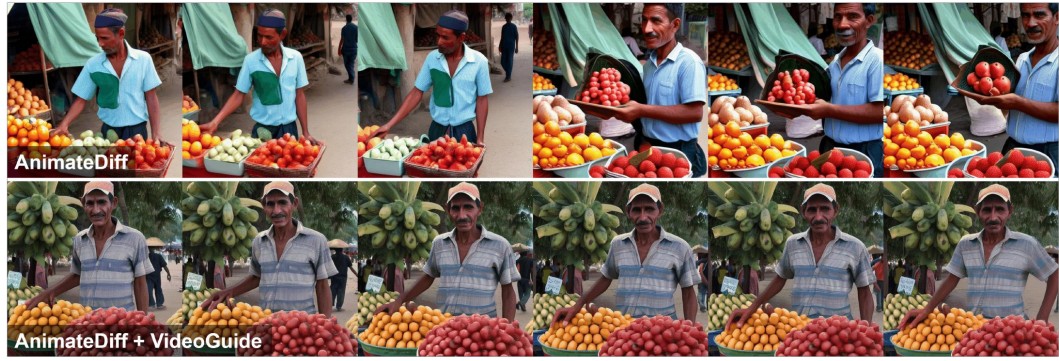

"A male vendor selling fruits"

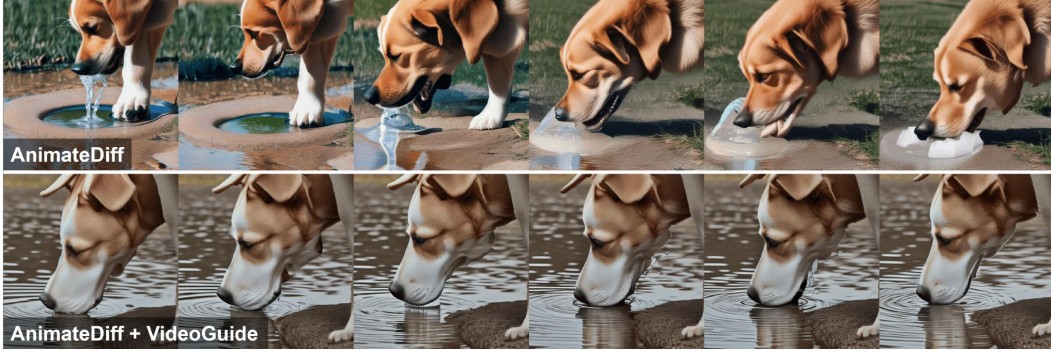

"A dog drinking water"

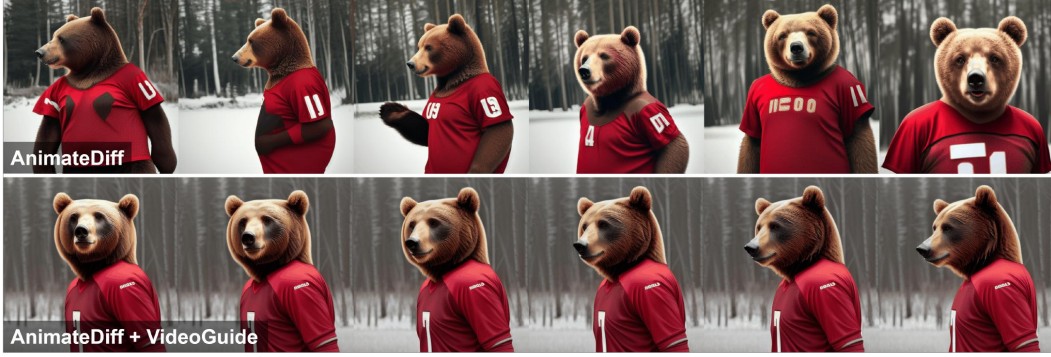

"A bear wearing red jersey"

Figure 8: More Qualitative Results of VideoGuide on AnimateDiff (with RealisticVision).

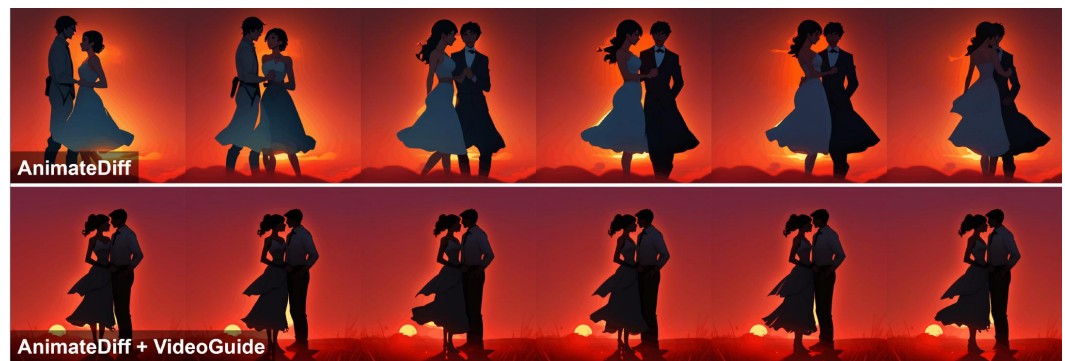

"Silhouette of the couple during sunset"

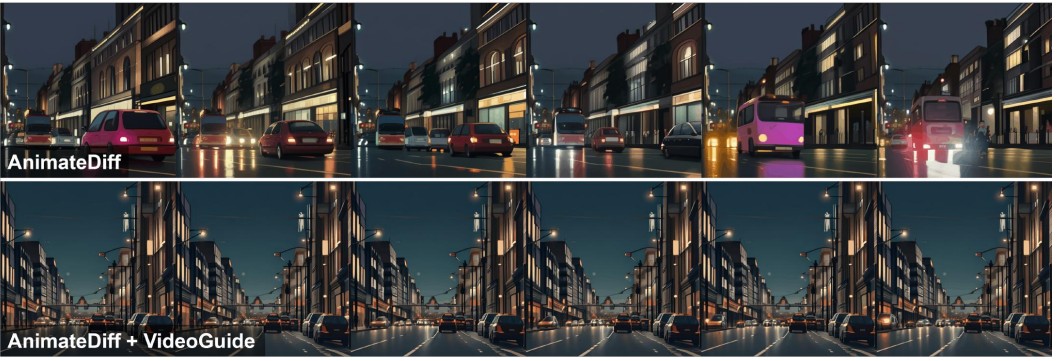

"Traffic in London street at night"

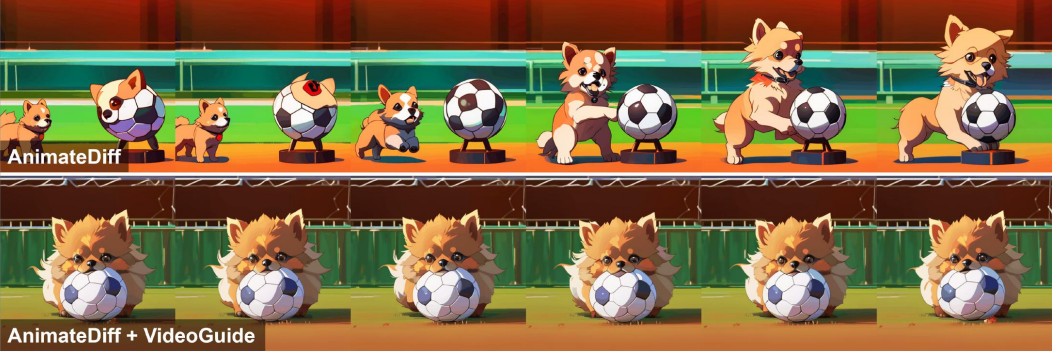

"A cute Pomeranian dog playing with a soccer ball"

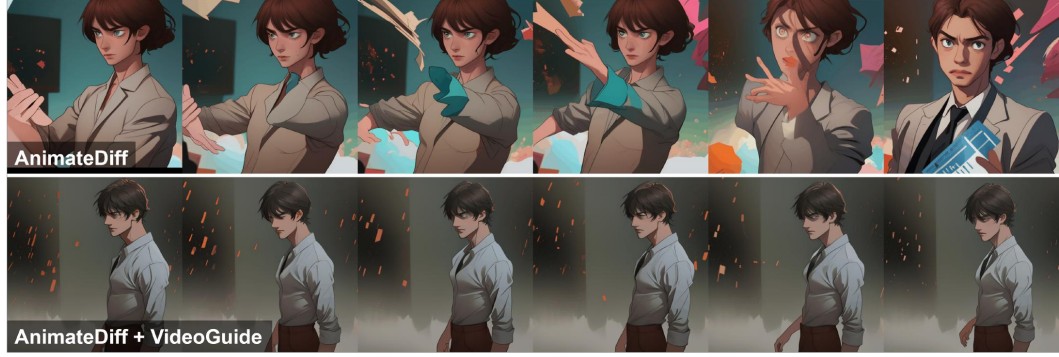

"A footage of actor movie scene"

Figure 9: More Qualitative Results of VideoGuide on AnimateDiff (with ToonYou).

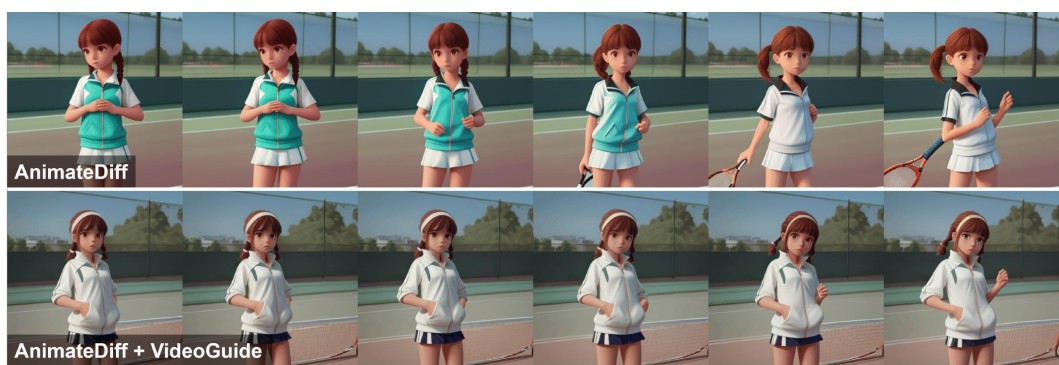

"A girl in her tennis sportswear"

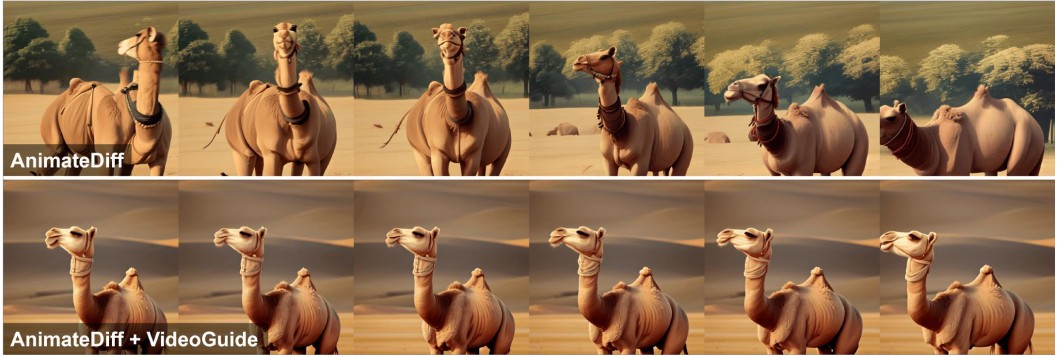

"Vertical video of camel roaming in the field during daytime"

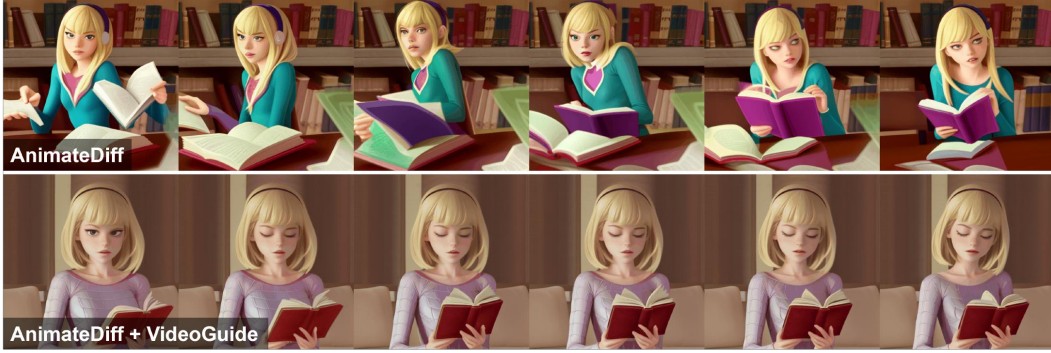

"Gwen Stacy reading a book"

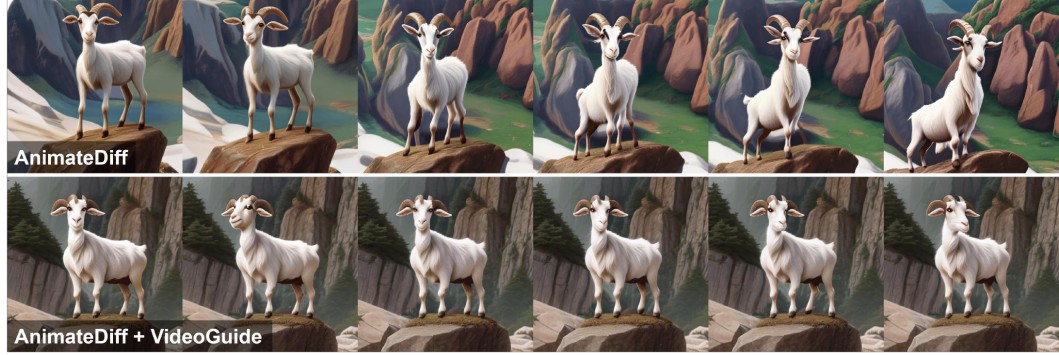

"Goat standing over a rock"

Figure 10: More Qualitative Results of VideoGuide on AnimateDiff (with RCNZCartoon).

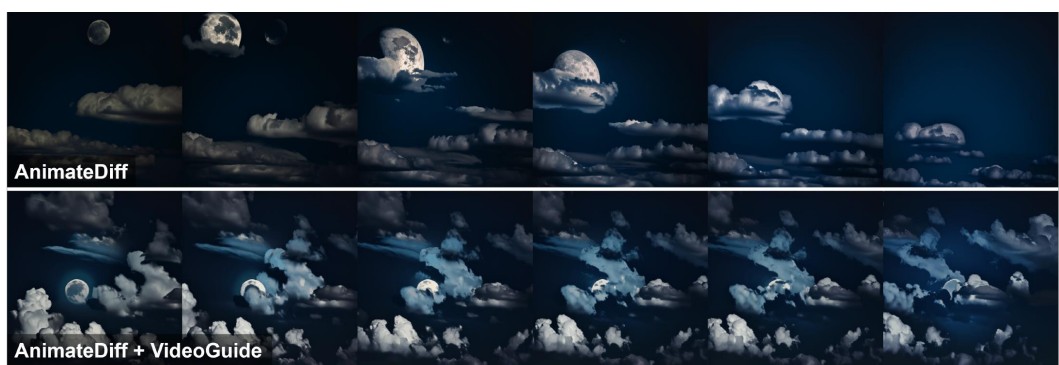

"Dark clouds overshadowing the full moon"

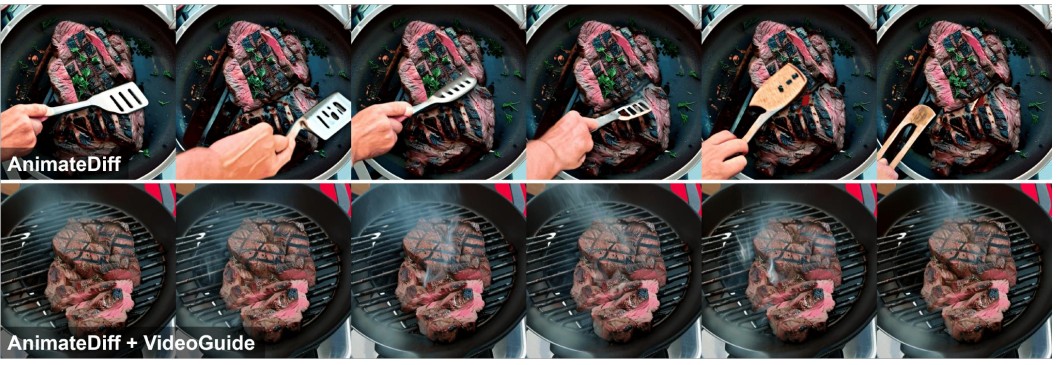

"Grilling a steak on a pan grill"

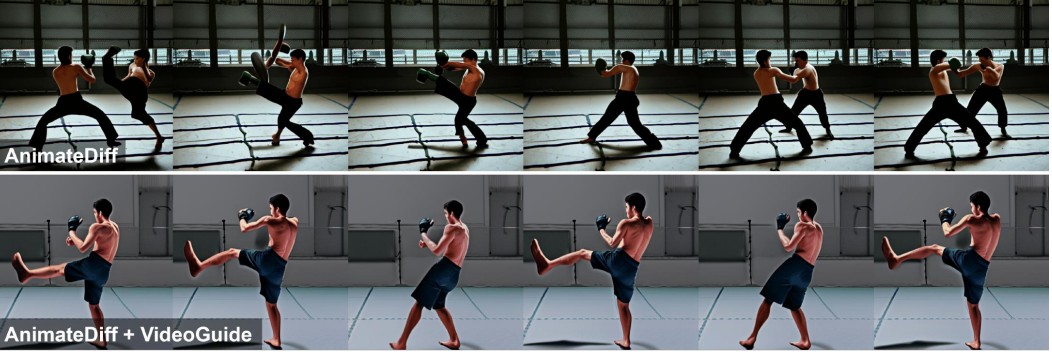

"Fighter practice kicking"

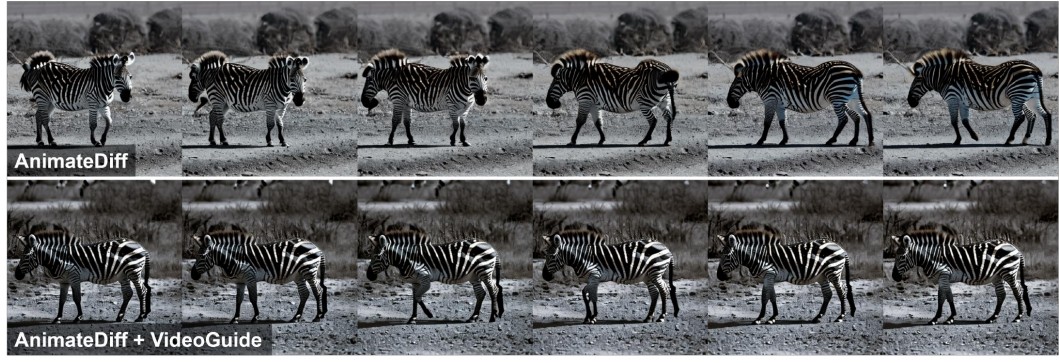

"A zebra taking a peaceful walk"

Figure 11: More Qualitative Results of VideoGuide on AnimateDiff (with FilmVelvia).

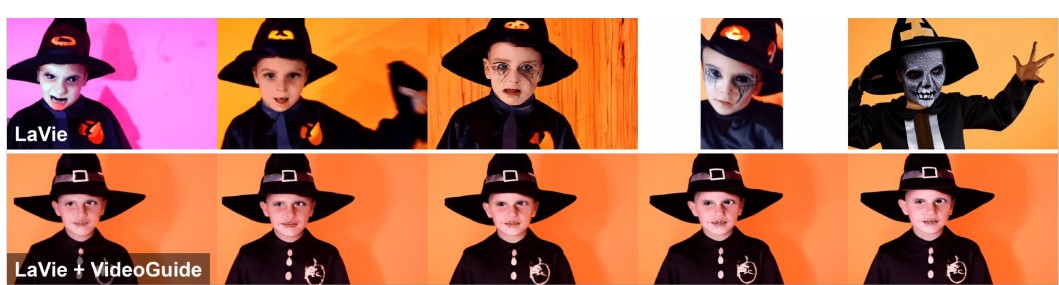

"Kid in a Halloween costume"

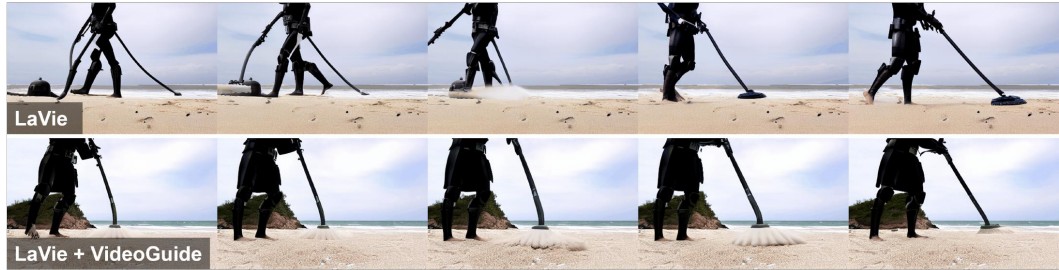

"A storm trooper vacuuming the beach"

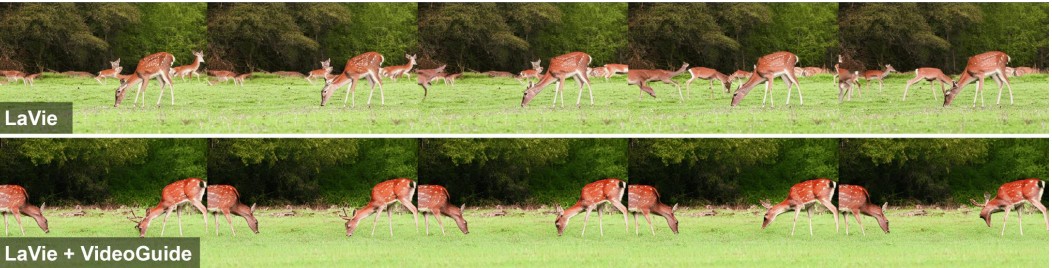

"Deer grazing in the field"

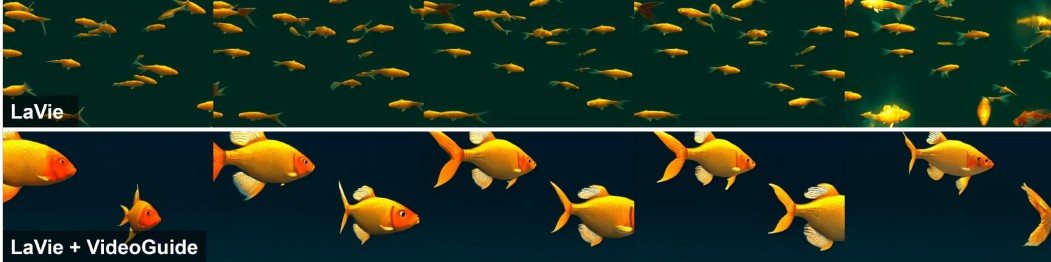

"Golden fish swimming in the ocean"

Figure 12: More Qualitative Results of VideoGuide on LaVie.

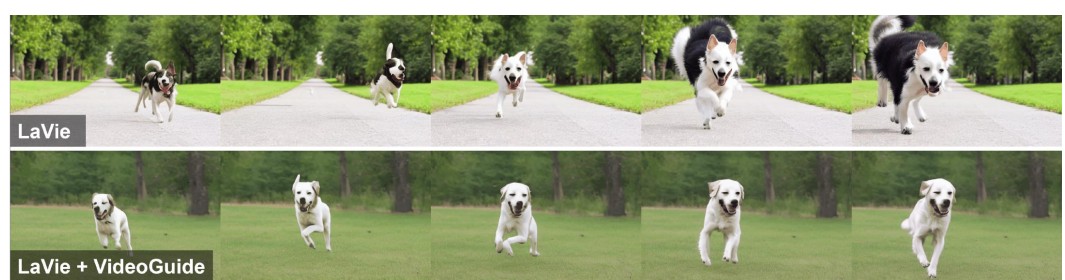

"A dog running happily"

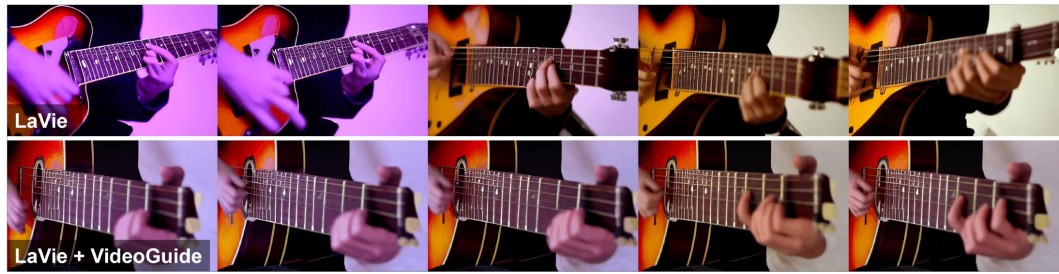

"A person playing guitar"

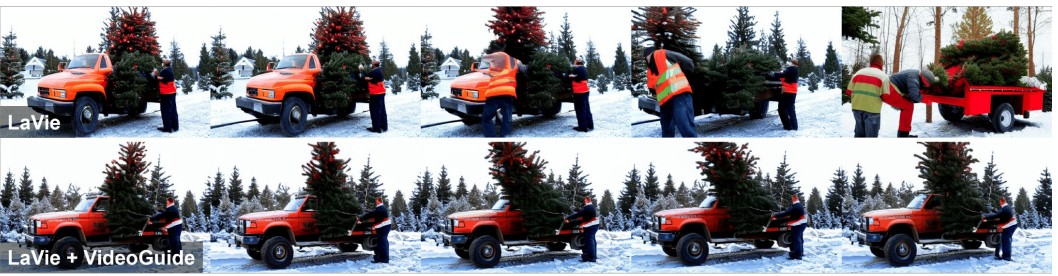

"Men loading Christmas tree on tow truck"

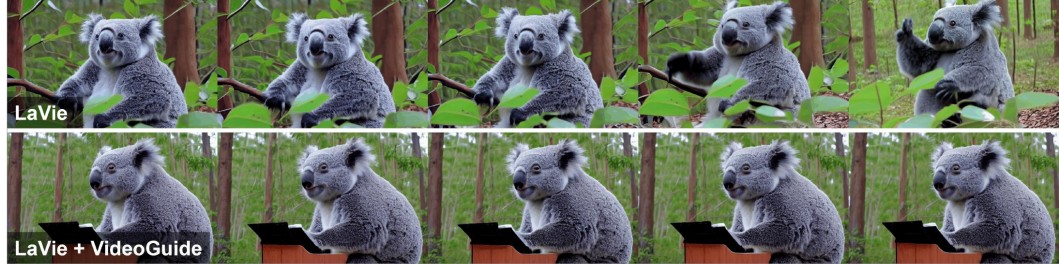

"A koala bear playing piano in the forest"

Figure 13: More Qualitative Results of VideoGuide on LaVie.

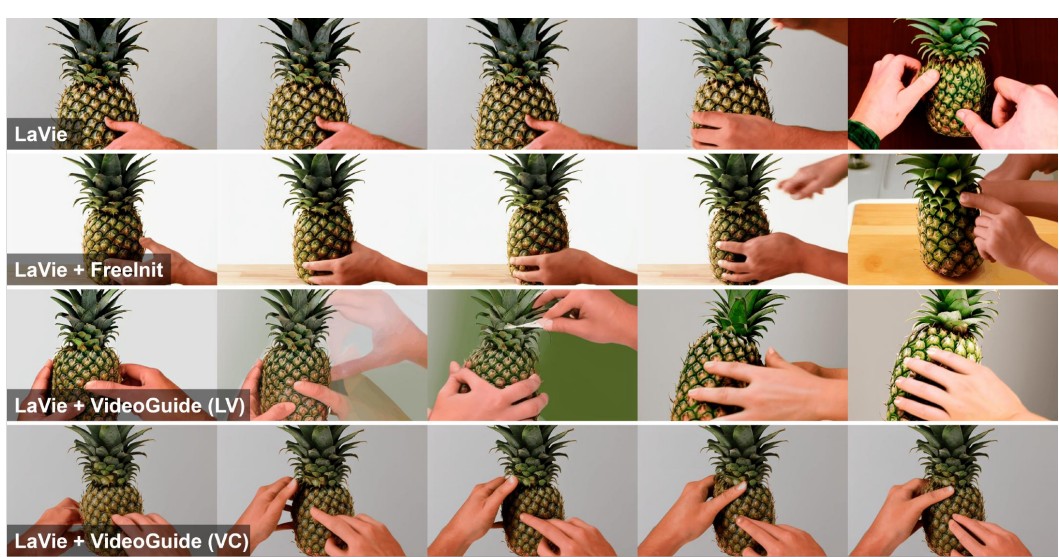

"Removing a pineapple leaf"

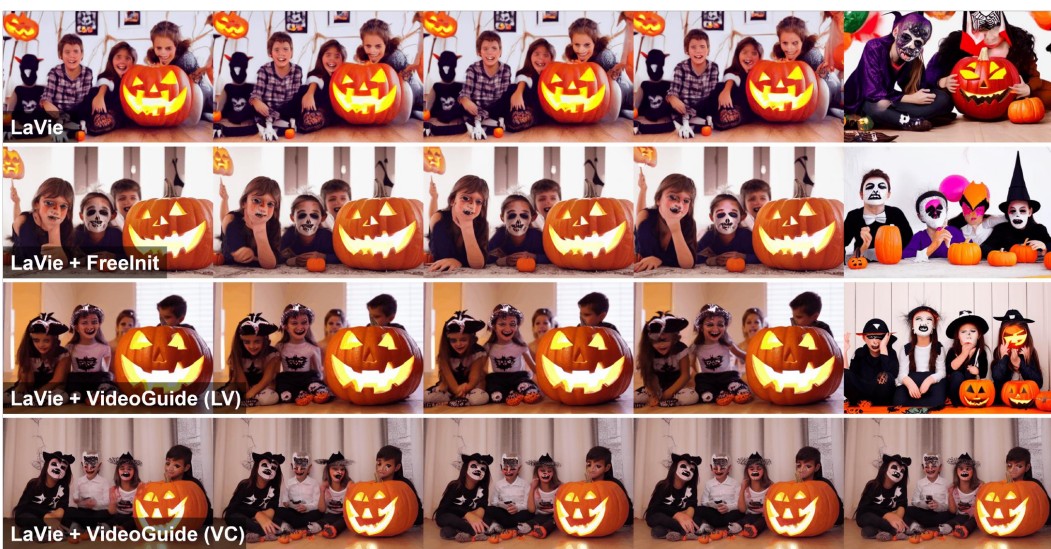

"Kids celebrating Halloween at home"

Figure 14: VideoGuide helps solve the issue of sudden frame shifts in LaVie samples. By integrating an external guiding model, VideoGuide provides smoother frame transitions to the base model. LV indicates that guidance model of LaVie is used (the self-guided case), and VC indicates that guidance model of VideoCrafter2 is used. Guidance given with the external model VideoCrafter2 solves sudden frame shift unsolvable by other methods.

