# OpenReview forum: "VideoGuide: Improving Video Diffusion Models without Training Through a Teacher's Guide"
_ICLR.cc/2025/Conference — ICLR 2025 Conference Withdrawn Submission_

### Official Review · Reviewer_ZaVE · 2024-10-29

**Soundness:** 2
**Presentation:** 3
**Contribution:** 2
**Rating:** 3
**Confidence:** 5

**Summary:**

This paper proposes a novel framework called VideoGuide, which enhances the temporal quality of pretrained text-to-video (T2V) diffusion models without requiring additional training or fine-tuning. VideoGuide leverages the strengths of any pretrained video diffusion model (VDM) to guide the denoising process of another VDM, improving temporal consistency and motion smoothness while maintaining imaging quality. The framework is versatile and can be applied to various base models, enabling underperforming models to benefit from the strengths of superior models.

**Strengths:**

1.	The framework can be applied to various base models, making it a versatile solution for improving temporal consistency and motion smoothness.
2.	VideoGuide demonstrates improved performance in terms of temporal consistency, motion smoothness, and imaging quality compared to prior methods.

**Weaknesses:**

1.	My primary concern about this paper is the limited novelty of the proposed method. Although the authors claim that their method uses teacher video diffusion models to guide existing video diffusion models, enhancing performance, essentially, the proposed method is just a combination of **multiple denoising-diffusion iterations** and high-low frequency decoupling operations, which is consistent with FreeInit, a validated effective strategy. Therefore, this paper does not provide sufficient contributions and further insights. As shown in Table 1, the improvement of the proposed method over FreeInit is not significant.
2.	Regarding the authors' claim that "the proposed method can use any pre-trained video diffusion model as guidance," they should provide more comprehensive arguments, which are not straightforward. The authors only introduce VideoCrafter2 as an external guidance, but it is unclear whether there are necessary constraints on the type of video diffusion models that can satisfy it as external guidance.
3.	Some necessary related works are missing in this paper, including UniCtrl [1] and I4VGen [2]. The authors need to provide necessary discussions in the paper.
4.	This paper primarily focuses on the temporal consistency and image quality of the generated videos. In fact, the motion degree of the synthesized videos is also equally important, as it is well-known that temporal consistency and motion degree are difficult to balance, and improving temporal consistency may compromise motion degree to some extent. VBench also provides corresponding metrics, and the authors need to provide necessary experimental results accordingly.
5.	Table 2 also provides the inference time of the video diffusion baseline.

[1] UniCtrl: Improving the Spatiotemporal Consistency of Text-to-Video Diffusion Models via Training-Free Unified Attention Control, Chen et al., arXiv 2024
[2] I4VGen: Image as Free Stepping Stone for Text-to-Video Generation, Guo et al., arXiv 2024

**Questions:**

In conclusion, considering the weaknesses mentioned above, this paper cannot be considered a well-prepared version for ICLR. Therefore, I lean towards rejecting this manuscript.

---

### Official Review · Reviewer_NUJc · 2024-10-31

**Soundness:** 2
**Presentation:** 2
**Contribution:** 3
**Rating:** 5
**Confidence:** 4

**Summary:**

this paper presents a training-free method to enhance the video generation performance by using the intermediate features from another model as a guidance. the guiding method is quite easy to implement as it is actually an interpolation of latents from sourcing and guiding models, when cfg++ and low-pass filter are applied. the experimental results demonstrate improved video generation quality with limited extra computation.

**Strengths:**

this paper presents an interesting idea of making low-performing models generate good results, or accelerating the computation of a high-performing models.

**Weaknesses:**

1. the experiments are not solid. (a) modern DiT-based video generation models should be included in the experiments. (b) the evaluation metrics may not correlate well with true performance of video generation models (human evaluations would be much better).
2. the writing needs a lot of improvement. for example, in the experiments section there is one sentence "we set I = 5, β = 0.5, and τ = 10", but the variable "I" was not introduced before.

**Questions:**

1. what will happen if CFG is used instead of CFG++? most of existing video generation models still use CFG, as CFG++ may not always perform better than CFG.
2. why is low-pass filter a must in your method? there lacks theoretical justification of doing so.
3. how do the evaluation metrics used in the paper correlate with the true performance of video generation models?

---

### Official Review · Reviewer_opCU · 2024-11-04

**Soundness:** 3
**Presentation:** 3
**Contribution:** 2
**Rating:** 5
**Confidence:** 5

**Summary:**

This paper proposed a training-free method for improving the temporal consistency of a video diffusion model. The motivation is that the predicted Z_0 at a smaller time step is more temporal consistent than the predicted Z_0 at a larger time step. More specifically, at each DDIM step, the Z_0 will be mixed together with a Z_0 at a smaller time step. Moreover,  the guiding model is not necessarily the same model as the sampling model. The idea has been validated on models including AnimateDiff and LaVie. The method has been proven effective using both quantitive numbers and user studies.

**Strengths:**

- The paper reads well and is easy to follow.
- The idea of mixing two Z_0 is interesting and effective.
- The proposed method has shown superiority over FreeInit in terms of both speed and quality.

**Weaknesses:**

- For the self-guidance scenario, does VideoCrafter2 show a similar improvement as AnimateDiff and LaVie? If not, then the benefit of this method might be saturate with a better video foundational model.
- It will be good to test the proposed idea on the latest DiT-based model such as OpenSora etc.
- In the scenario of using a different model as the guidance model, why not directly use the guidance model without the proposed idea? Other than personalization, the paper mentioned super-resolution and interpolation models of Lavie, but they can also be used together with other T2V foundational models.
- It will be good to add the temporal consistency and dynamic degree metric from VBench. The proposed idea might have worse performance on the dynamic degree metric.

**Questions:**

- Are the video results of self-guidance included in the project page?
- Can the guidance model have a different VAE from the sampling model?
- Any ablation results on the frequency filtering component?

---

### Official Review · Reviewer_yfFm · 2024-11-04

**Soundness:** 2
**Presentation:** 2
**Contribution:** 2
**Rating:** 3
**Confidence:** 5

**Summary:**

This work introduces a framework called VideoGuide aimed at enhancing the temporal consistency of pretrained text-to-video (T2V) diffusion models without the need for additional training or fine-tuning. The authors achieve this by leveraging any pretrained video diffusion model (VDM) or the model itself as a guide during the early stages of inference, improving temporal quality through the interpolation of the guiding model's denoised samples into the sampling model's denoising process. This method not only significantly improves temporal consistency but also maintains image quality, providing a cost-effective and practical solution.

**Strengths:**

VideoGuide combines existing diffusion models with a novel interpolation technique to enhance temporal consistency without additional training. This approach leverages the strengths of different models, such as the temporal consistency of Videocrafter2 and the personalization capabilities of AnimateDiff.

The paper provides extensive experimental validation using multiple state-of-the-art T2V diffusion models, including Videocrafter and AnimateDiff. The authors conduct thorough evaluations using both quantitative metrics (e.g., subject consistency, background consistency, imaging quality, and motion smoothness) and qualitative user studies. The results consistently show significant improvements in temporal consistency and overall video quality, demonstrating the robustness and effectiveness of the proposed method.

**Weaknesses:**

The paper does not provide a clear and detailed explanation of why the proposed interpolation method works to improve temporal consistency. This lack of theoretical justification can leave readers questioning the validity and reliability of the method.
 Without a solid theoretical foundation, the method may appear ad-hoc, and readers may find it difficult to understand the underlying mechanisms that contribute to its effectiveness.

The paper lacks visualizations of the sampling trajectories, which are crucial for providing an intuitive understanding of the method's behavior. Such visualizations could help readers see how the interpolation process affects the temporal consistency of the generated videos. The absence of these visualizations makes it harder for readers to grasp the practical implications of the method and to verify the claimed improvements in temporal consistency.

 The paper does not adequately discuss the relationship and differences between the proposed method and previous approaches aimed at improving temporal consistency in video generation. There is a lack of detailed comparisons that highlight the unique contributions and advantages of VideoGuide. Without a thorough comparison, readers may not fully appreciate the novelty and significance of the proposed method. This can undermine the perceived value and impact of the research.

**Questions:**

The paper does not provide a clear and detailed explanation of why the proposed interpolation method works to improve temporal consistency. This lack of theoretical justification can leave readers questioning the validity and reliability of the method.
 Without a solid theoretical foundation, the method may appear ad-hoc, and readers may find it difficult to understand the underlying mechanisms that contribute to its effectiveness.

The paper lacks visualizations of the sampling trajectories, which are crucial for providing an intuitive understanding of the method's behavior. Such visualizations could help readers see how the interpolation process affects the temporal consistency of the generated videos. The absence of these visualizations makes it harder for readers to grasp the practical implications of the method and to verify the claimed improvements in temporal consistency.

 The paper does not adequately discuss the relationship and differences between the proposed method and previous approaches aimed at improving temporal consistency in video generation. There is a lack of detailed comparisons that highlight the unique contributions and advantages of VideoGuide. Without a thorough comparison, readers may not fully appreciate the novelty and significance of the proposed method. This can undermine the perceived value and impact of the research.

---

### Note · Authors · 2024-11-13

I have read and agree with the venue's withdrawal policy on behalf of myself and my co-authors.